# Apelin Resistance Contributes to Muscle Loss during Cancer Cachexia in Mice

**DOI:** 10.3390/cancers14071814

**Published:** 2022-04-02

**Authors:** Andrea David Re Cecconi, Mara Barone, Mara Forti, Martina Lunardi, Alfredo Cagnotto, Mario Salmona, Davide Olivari, Lorena Zentilin, Andrea Resovi, Perla Persichitti, Dorina Belotti, Federica Palo, Nobuyuki Takakura, Hiroyasu Kidoya, Rosanna Piccirillo

**Affiliations:** 1Department of Neurosciences, Istituto di Ricerche Farmacologiche Mario Negri IRCCS, Via Mario Negri 2, 20156 Milan, Italy; andrea.rececconi@marionegri.it (A.D.R.C.); mara.barone@marionegri.it (M.B.); forti9410@gmail.com (M.F.); martina.lunardi@guest.marionegri.it (M.L.); davide.olivari@marionegri.it (D.O.); federica.palo@guest.marionegri.it (F.P.); 2Molecular Biochemistry and Pharmacology Department, Istituto di Ricerche Farmacologiche Mario Negri IRCCS, Via Mario Negri 2, 20156 Milan, Italy; alfredo.cagnotto@marionegri.it (A.C.); mario.salmona@marionegri.it (M.S.); 3Molecular Medicine, International Centre for Genetic Engineering and Biotechnology, Via Padriciano 99, 34149 Trieste, Italy; lorena@icgeb.org; 4Department of Oncology, Istituto di Ricerche Farmacologiche Mario Negri IRCCS, Via Stezzano 87, 24126 Bergamo, Italy; andrea.resovi@marionegri.it (A.R.); perla.persichitti@marionegri.it (P.P.); dorina.belotti@marionegri.it (D.B.); 5Department of Signal Transduction, Research Institute for Microbial Diseases, Osaka University, 3-1 Yamada-oka, Suita-shi, Osaka 565-0871, Japan; ntakaku@biken.osaka-u.ac.jp; 6Department of Integrative Vascular Biology, Faculty of Medical Sciences, University of Fukui, 23-3 Matsuoka-Shimoaizuki, Eiheiji, Yoshida, Fukui 910-1193, Japan; kidoya@u-fukui.ac.jp

**Keywords:** cancer cachexia, apelin, muscle wasting, hyperapelinemia

## Abstract

**Simple Summary:**

Cancer cachexia is a highly debilitating syndrome involving severe body weight loss. Worldwide around 9–14.5 million cancer patients suffer from cachexia every year and many of them die because of cachexia. Our study aimed to assess the possible role of apelin against muscle loss during cancer growth given its beneficial effects against muscle atrophy during aging. We found apelin exhibiting advantageous effects against atrophy in in vitro models, but not in in vivo models, where we unraveled undesirable apelin resistance that may nullify apelin-based therapy for cancer cachexia.

**Abstract:**

Cancer cachexia consists of dramatic body weight loss with rapid muscle depletion due to imbalanced protein homeostasis. We found that the mRNA levels of apelin decrease in muscles from cachectic hepatoma-bearing rats and three mouse models of cachexia. Furthermore, *apelin* expression inversely correlates with *MuRF1* in muscle biopsies from cancer patients. To shed light on the possible role of apelin in cachexia in vivo, we generated apelin 13 carrying all the last 13 amino acids of apelin in D isomers, ultimately extending plasma stability. Notably, apelin D-peptides alter cAMP-based signaling in vitro as the L-peptides, supporting receptor binding. In vitro apelin 13 protects myotube diameter from dexamethasone-induced atrophy, restrains rates of degradation of long-lived proteins and *MuRF1* expression, but fails to protect mice from atrophy. D-apelin 13 given intraperitoneally for 13 days in colon adenocarcinoma C26-bearing mice does not reduce catabolic pathways in muscles, as it does in vitro. Puzzlingly, the levels of circulating apelin seemingly deriving from cachexia-inducing tumors, increase in murine plasma during cachexia. Muscle electroporation of a plasmid expressing its receptor APJ, unlike apelin, preserves myofiber area from C26-induced atrophy, supporting apelin resistance in vivo. Altogether, we believe that during cachexia apelin resistance occurs, contributing to muscle wasting and nullifying any possible peptide-based treatment.

## 1. Introduction

Cancer cachexia is a highly debilitating syndrome involving severe body weight loss, mainly due to muscle mass depletion with or without fat tissue loss [1,2]. It afflicts up to 80% of patients with advanced cancers [3]. Remarkably, the chronic loss of muscle mass causes functional impairment, fatigue, and cardio-respiratory complications, ultimately leading to death in 20–48% of cases [3]. Its prevention could thus boost the survival of cancer patients, as we [4] and others [5] have shown in mice. There are still no effective treatments for muscle wasting associated with cancer.

Muscle atrophy during cancer is the result of imbalanced protein homeostasis. Indeed, protein catabolism is highly enhanced and driven by muscle-specific ubiquitin ligases as MuRF1 and atrogin-1 [2]. Systemic inflammation too plays a role in the progression of cancer cachexia: proinflammatory cytokines, such as IL-6 or TNF-α, are elevated in some cachectic patients’ plasma and cancer-bearing models [3]. Furthermore, insulin resistance occurs during cancer cachexia and has been described in *Drosophila Melanogaster* [6,7], in mice [8] as in humans [9]. Insulin is an anabolic factor that counteracts protein degradation and promotes protein synthesis in muscles, so insulin insensitivity further aggravates muscle loss with cachexia [2].

To identify genes implicated in muscle wasting typical of cancer, but not because of other conditions, such as uremia, disuse, fasting, or diabetes, we re-analyzed previous gene expression profiles generated by microarrays [10]. We found, among others [11], the gene encoding for *apelin* specifically and drastically downregulated in muscles of only Yoshida hepatoma-bearing rats.

Apelin is a muscle-, fat- and brain-secreted factor, which has cardio-protective, anti-obesity, and anti-diabetic properties [12,13]. This last function is due to the fact that it can alter glucose signaling, similarly to insulin [14] and is necessary for the maintenance of insulin sensitivity [15]. Apelin is a regulatory peptide that can bind the apelin receptor, namely angiotensin-like-receptor 1 (APJ) [16], a member of G protein-coupled receptors (GPCRs). Apelin and APJ—which is expressed also in human and murine skeletal muscles [17]—are widely distributed in the body and are involved in various physiological functions as blood pressure regulation [18]. Apelin and APJ mRNAs are also enhanced in cancerous tissues and thought to play a role in promoting angiogenesis during cancer progression [19]. Depending on the cell type studied, APJ activation enhances extracellular signal-regulated kinases, AMPK, AKT, and p70S6 kinases and the inhibition of cyclic AMP (cAMP) production [20]. Activation of these pathways would result in worthwhile increases in the protein content of skeletal muscles. Apelin has in fact proved beneficial in muscle and counteracts age-associated atrophy (i.e., sarcopenia) [21].

The *apelin* gene encodes for a 77 amino acid-long preproapelin that generates several molecular isoforms, such as apelin 36, 17, and 13 by post-translational processing [22], but have a short half-life in vivo [23]. The last 13 amino acids of apelin (apelin-13) seem sufficient to bind with nanomolar affinity APJ and exert their biological function [16,24,25] also in muscles. To manipulate this pathway in mice, we tested, in muscles in vivo, plasmids or adeno-associated viruses expressing preproapelin (or its receptor). Furthermore, we generated de-novo apelin 13-based peptides where all the amino acids appear as D isomers and evaluated them against cachexia in mice. These peptides had longer half-lives in vivo, while still retaining receptor binding ability.

Through in silico, in vitro, and in vivo approaches and providing data on *apelin* expression in three distinct species (*Rattus Norvegicus*, *Mus Musculus*, and *Homo sapiens*), we noted apelin resistance in cancer cachexia and found that it may further exacerbate muscle loss.

## 2. Materials and Methods

### 2.1. Cell Culture

C2C12 (ATCC, Manassas, VA, USA), a myoblast cell line from the C3H mouse strain, was grown in DMEM (Dulbecco’s Modified Eagle’s Medium, Gibco, Waltham, MA, USA) with fetal bovine serum (FBS) (Euroclone, Pero, Italy) and 2 mM L-glutamine, (BioWest, Nuaillè, France) and maintained in culture at 37 °C with 5% CO_2_. Myoblasts differentiate into myotubes when they reached 80% confluence and were cultured for four days in DMEM, 2 mM L-glutamine, horse serum (HS) (Euroclone), at 37 °C and 8% CO_2_. The differentiation medium was changed every two days. C26 and LLC cells were grown in DMEM with 10% FBS and 2 mM L-glutamine at 37 °C with 5% CO_2_. 4T1 cells were grown in Ham’s F12 (Gibco) with 10% FBS and 2 mM L-glutamine at 37 °C with 5% CO_2_.

C26 cells were donated by Prof. Mario Paolo Colombo (IRCCS-Istituto Nazionale dei Tumori, Milan, Italy). LLC cells were donated by Prof. Paola Costelli (University of Turin, Turin, Italy). MCG101 is a sarcoma cell line, grown in McCoy’s 5A (Gibco) medium with 10% FBS and 2 mM L-glutamine at 37 °C with 5% CO_2_. MCG101 cells were shared by Prof. Anders Blomqvist (Linköping University, Linköping, Sweden). The FC1199 pancreatic cancer cell line, derived from tumors arisen in *LSL-Kras^G12D^*^/*+*^*;LSL-Trp53^R172H^*^/*+*^*;Pdx-1-Cre* mice in the C57BL/6 background, was provided by Prof. D.A. Tuveson (Cold Spring Harbor, New York, NY, USA). Cells were cultured in DMEM supplemented with 10% fetal calf serum (FCS) (Euroclone) and 1% L-glutamine [26]. Human myoblasts cell line (kind gift from Prof. Benedikt Schoser, from Germany) was grown in Skeletal Muscle Cell Growth Medium with Supplement Mix (PromoCell, Heidelberg, Germany) and maintained in culture at 37 °C with 5% CO_2_. Myoblasts differentiate into myotubes when they reached 80% confluence and were cultured for four days in Skeletal Muscle Differentiation Medium with Supplement Mix (PromoCell) at 37 °C and 8% CO_2_. The differentiation medium was changed every two days. All cells used were not contaminated by mycoplasma.

### 2.2. Adeno-Associated Vectors, Plasmids, and Drugs

pEGFP-N1-APJ and pCAG-murine preproapelin plasmids were described in [27] and [28], respectively, and the latter was cloned into a plasmid bearing an adenoassociated virus (AAV) backbone. pLKO plasmid was purchased from Open Biosystem. In luciferase assays, we used as plasmids mMURF_pGL4.10 LUC2 generated by us in collaboration with Dr. Rizzi from University of Milan, Milan, Italy, pGL4.29 luc2P/CRE/Hygro (Promega, Madison, WI, USA) and pRL-TK Renilla (Promega). We used 10 nM [Pyr1]-Apelin 13 (Peptide institute, INC, Osaka, Japan), 10 nM Apelin 36 (Tocris Biosciences Bristol, UK), or 1 or 10 μM Dexamethasone (Sigma, St. Louis, MO, USA) and 50 µM forskolin (Sigma). Recombinant AAV vectors were packaged into AAV capsid serotype 9 by the AAV Vector Unit at the International Centre for Genetic Engineering and Biotechnology in Trieste, as previously described [29,30]. Titers obtained were in the range of 1 × 10^12^ to 1 × 10^13^ vg per mL.

### 2.3. Protein Degradation Assay

C2C12 myotubes expressing pLKO or preproapelin-based plasmids for at least 16 h were incubated with 3H-tyrosine (2 μCi/mL; PerkinElmer, Waltham, MA, USA) for 24 h to label long-lived proteins and then processed as in [11].

### 2.4. Diameter Measurements

Pictures of myotubes were acquired with an Olympus Microscope IX71 (10× magnification, 10× ocular lens, Olympus, Shinjuku, Japan) with Cell F (2.6 Build1210, Olympus) imaging software for Life Science microscopy (Olympus Soft Imaging solution GmbH, Munster, Germany). Myotube diameters were measured in blind conditions using ImageJ software (National Institutes of Health, Bethesda, MA, USA). For each photo 10–11 myotubes were measured, calculating the mean of three diameters per myotube, for a total of at least 100 cells per condition.

### 2.5. Luciferase-Based Assays

We used C2C12 myoblasts grown in a 24-well plate and transfected with pLKO or preproapelin-expressing plasmids with murine MuRF1 promoter-FLuc plasmid, and pRL-TK plasmid (Promega) using Lipofectamine 2000 (Life Technologies Europe BV, Carlsbad, CA, USA). The next day, they were treated with vehicle or 10 μM dexamethasone for 24 h. Luciferase activities were measured using the Dual-Luciferase Reporter Assay System (Promega) and a luminometer (Glomax 20/20 single tube luminometer, Promega).

### 2.6. Apelin Peptide Synthesis

D-apelin 13 was synthesized by solid-phase chemistry using fluorenylmethyloxycarbonyl chloride group (Fmoc) protected d-amino acid, with Initiator+Alstra peptide synthesizer (Biotage, Upsala, Sweden) at 0.1 mM scale on Rink-amide resin. Fmoc deprotection was done automatically at RT by treating the peptide-resin with 20% piperidine in DMF (Dimethylformammide) for 3 min, followed by another cycle of 10 min and 4 times DMF-washed. Amino acids were activated using DIC (N,N′-Diisopropylcarbodiimide) and Oxyma pure, both at 0.5 mM in DMF. D-apelin 13 peptide was cleaved from the resin with TFA (trifluoroacetic acid): triisopropylsilane solution (95:5 *v/v*), precipitated and washed with cold diethyl ether. The crude peptide was purified in reverse phase HPLC using a semi-preparative C4 column (Symmetry 300, Waters, Milford, MA, USA), with mobile phases of 0.1% TFA in water (eluent A)/0.08% TFA in acetonitrile (eluent B) and a linear gradient from 5 up to 100% of eluent B in 60 min. The peaks were collected and characterized by matrix-assisted laser desorption/ionization-time-of-flight (MALDI-TOF) mass spectrometry using an ABI 4800 mass spectrometer, operating in reflector mode. Solutions containing D-apelin 13 peptide with a purity greater than 95% were finally frozen, dried, and the powders stored at −20 °C until use. The d-Fmoc-amino acids were obtained from ANASPEC (Fremont, CA, USA), the other reagents used for peptide synthesis were obtained from Sigma-Aldrich company.

### 2.7. RNA Isolation from Cultured Cells or Muscles and Reverse Transcription

Total RNA was isolated from cells or muscles with QIAzol Lysis Reagent (Qiagen, Hilden, Germany) and miRNeasy Kit (Qiagen), as in [31]. RNA concentration and purity were measured in a spectrophotometer (NanoDrop 1000, ThermoFisher Scientific, Waltham, MA, USA).

### 2.8. Quantitative Real-Time Polymerase Chain Reaction (PCR)

mRNA/μg in muscle was analyzed using TaqMan reverse transcription reagents (Life Technologies) or SYBR Green (Qiagen). *β-Glucuronidase* (*GUSB*), or *Tata binding protein* (*TBP*), or *Importin 8* (*IPO8*), or *18S ribosomal RNA* (*18S*) were used as housekeeping genes (HK). To normalize for three HK, we used the ratio between the arithmetic average and the geometric average of the ng of cDNA of each gene, as described in [32]. All the primers used in this study are listed in Appendix A.

### 2.9. Protein Extraction and Western Blot

Total proteins were extracted from myotubes and muscles and quantitated as in [31]. Proteins were separated by electrophoresis on 4–20% sodium dodecyl sulfate-polyacrylamide gel electrophoresis (SDS-PAGE) (Bio-Rad, Hercules, CA, USA) and transferred to a polyvinylidene difluoride membrane (GE Healthcare, Chicago, IL, USA), which was then saturated for 2 h at RT with 5% bovine serum albumin (BSA) or milk in TBS-T buffer (20 mM Tris, 150 mM NaCl and 0.1% Tween-20) (Sigma). The membrane was then incubated with the primary antibody O/N at 4 °C. The following primary antibodies were used: anti-AKT (BK9272s, Cell Signaling, Danvers, MA, USA), anti phospho-Ser473AKT (BK9271S, Cell Signaling), anti-4EBP1 (9452, Cell Signaling), anti-phosphoSer65-4EBP1 (9451, Cell Signaling), anti-vinculin (V9264, Sigma), anti-apelin (PA1501, Boster Bio, Pleasanton, CA, USA), anti-APJ (ABD43, Millipore, Burlington, MA, USA), anti-phosphoSer65-4EBP1 (9451, Cell Signaling), anti-4EBP1 (9452, Cell Signaling), anti-phosphoThr172-AMPKα (2535, Cell Signaling), anti-AMPKα (2603, Cell Signaling), anti-phosphoSer2448mTOR (2971, Cell Signaling), anti-mTOR (2972, Cell Signaling) and anti-GAPDH (G8795, Sigma). Anti-MuRF1 antibodies were donated by Prof. Alfred L. Goldberg (Harvard Medical School, Boston, MA, USA). Secondary antibodies were conjugated to alkaline phosphatase (Promega). Band intensities were analyzed using ImageJ software.

### 2.10. Enzyme-Linked Immunosorbent Assay (ELISA)

According to the manufacturer’s protocol, apelin levels in murine plasma were measured in ELISA (CSB-E12027m, Cusabio, Houston, TX, USA). Values below the detection limit were not included in the analyses and related graphs. The detection range is 31.25 pg/mL–2000 pg/mL.

### 2.11. MALDI-TOF Analysis

The stability of apelin 13 peptide was investigated using a MALDI-TOF 4800 mass spectrometer (Thermofisher Scientific). [Pyr1]-Apelin 13 (0.5 µg/µL) was added to mouse serum and 10 µL of the sample were taken at different times and mixed with 20 µL of a solution containing acetonitrile and 3% trifluoroacetic acid. After 10 min of incubation on ice with stirring and 3 min of centrifugation at 4000 rpm, 1 µL of the solution was used for MALDI-TOF analysis, in reflector mode using an HCCA matrix (α-cyano-4-hydroxycinnamic acid).

### 2.12. In Vivo Animal Models

C26 (10^6^ cells) or 4T1 (2 × 10^5^) were injected subcutaneously into the upper right flank of BALB/c mice (Envigo, Indianapolis, IN, USA). MCG101 (0.5 × 10^6^) or LLC cells (10^6^ cells) were injected subcutaneously in C57BL6/J mice (Envigo). Mice (at least five/group) were randomly allocated to different groups based on body weights and sacrificed when showing any clear signs of distress or 14 days after tumor transplant, when they were electroporated as in [31]. All animals were 9–10 week-old males at the time of tumor injection. Male mice were used throughout all the study, to avoid confounding factors (i.e., various phases of the estrous cycle). To minimize the differences in starting body and muscles weights of animals tested, we used mice with very similar body weights ± 0.5 g, so that at the time of tumor injection, less variability could be present. Tumors were measured manually with a caliper three times a week. Mice were considered cachectic when exhibiting significant muscle loss with respect to cancer-free mice that, at the time of injection, had comparable body weights to cancerous mice. FC1199 cells (1 × 10^4^ and 5 × 10^3^ cells) were implanted orthotopically in the pancreas as described [26,33,34] and mice sacrificed after 26 and 31 days, respectively. Cachexia was evaluated as TA weight loss at the end of the experiment.

In accordance with institutional guidelines, animals were killed when at least four out of five signs of distress (inactivity, kyphosis, ruffled fur, low body temperature, tremor) were present or when >20% of body weight was lost in consecutive 72 h. Procedures involving animals and their care were conducted in conformity with institutional guidelines in compliance with national and international laws and policies. The IRFMN adheres to the principles set out in the following laws, regulations, and policies governing the care and use of laboratory animals: Italian Governing Law (D.lgs 26/2014; Authorization No. 19/2008—A issued 6 March 2008 by Ministry of Health); Mario Negri Institutional Regulations and Policies providing internal authorization for persons conducting animal experiments (Quality Management System Certificate—UNI EN ISO 9001:2015—Reg. No. 6121); the NIH Guide for the Care and Use of Laboratory Animals (2011 edition), and EU directives and guidelines (EEC Council Directive 2010/63/UE). The Statement of Compliance (Assurance) with the Public Health Service (PHS) Policy on Human Care and Use of Laboratory Animals has been reviewed (Animal Welfare Assurance #294/2018-PR and #519/2021-PR). All animal protocols conform to the Guide for the Care and Use of Laboratory Animals generated by the Institute for Laboratory Animal Research, National Research Council of the National Academies.

### 2.13. Muscle Sample Processing and Fiber Size Measurements

Pictures of fibers of ten-μm thick sections of electroporated muscles were acquired with an Olympus Microscope IX71 (20× magnification, 10× ocular lens, Olympus, Shinjuku, Japan) with Cell F (2.6Build1210, Olympus) imaging software for Life Science microscopy (Olympus Soft Imaging solution GmbH, Munster, Germany). Cross-sectional areas (CSA) of fibers from the same muscle were measured in blind conditions using ImageJ software.

### 2.14. Patients Data Analysis

Rectus abdominis muscles were harvested by the University of Calgary Hepatobiliary/Gastrointestinal Tumor Bank at the time of laparotomy from patients with abdominal malignancies [35]. Clinical characteristics of the patients are described in [35]. For sample processing and microarray analysis, please refer to [35].

### 2.15. Sulforhodamine B (SRB) Assay

The viability of tumoral cells (LLC, MCG101 and C26) treated for 24 or 48 h with L- or D-apelin 13, ranging from 1 to 250 µM was measured with sulforhodamine B (SRB) assay (Sigma), using water as vehicle.

### 2.16. Statistical Analysis

The sample size was determined by power analysis with G*Power. All the experiments were repeated at least twice. For statistical analysis, data (means ± standard errors of the mean or SEMs) were analyzed with GraphPad Prism 8 for Windows (Graph-Pad Software, San Diego, CA, USA) and Statview Software for Windows (SAS StatView for Windows Redmond, WA, USA), with the following statistical tests: ordinary one-way or two-way analysis of variance (ANOVA) for multiple comparisons followed by Tukey’s or Dunnett’s post hoc test; Kruskal–Wallis followed by Dunn’s multiple comparison test. Unpaired *t*-test or Mann-Whitney test was used for comparisons of two groups. Spearman’s test was used for correlations. *p*-values ≤ 0.05 were considered significant.

## 3. Results

### 3.1. The Expression of Apelin Is Low in Cachectic Muscles of Rodents and Patients with Cancer

To identify genes specifically modulated in cancer cachexia, we interrogated a previously generated mouse microarray data set (cDNA hybridized against mouse GEM1 microarray) and identified the genes upregulated (-fold change > 2) or downregulated (-fold change ≤ 0.5) (*p* ≤ 0.05) only in the cachectic gastrocnemius of rats bearing Yoshida hepatoma for five days and not in any of the other muscle atrophy conditions tested (uremia, disuse, fasting, diabetes) (*p* ≤ 0.05). *Apelin* was among the most downregulated genes in hepatoma-bearing rats: its expression was strikingly reduced to 8% of controls.

To validate this in another species, we tested *apelin* expression in mice suffering from cachexia because of one of these tumors: colon adenocarcinoma (C26), methylcholanthrene-induced sarcoma (MCG101), or Lewis Lung carcinoma (LLC). When inoculated subcutaneously in mice, all these tumors cause rapid body weight loss (BWL) to different extents (Figure 1A–C). Although the tumors grew with different rates (Figure 1D), they all caused depletion of muscles, as indicated by reduced weight of gastrocnemius (Figure 1E) and tibialis anterior (TA) (Figure 1F). Subcutaneous growth of the tumor C26 caused the most severe TA atrophy—around 50% of controls, while MCG101 and LLC caused less atrophy (up to 20% of muscle weight loss, MWL) (Figure 1E,F).

In the cachectic TA of mice bearing C26, MCG101, or LLC, *MuRF1*, a known marker of muscle wasting in animal models, was induced about 3-, 8- and 2-folds, respectively (Figure 2A–C). Similar inductions were found for other ubiquitin ligases involved in muscle wasting, as *atrogin-1* [36] and muscle ubiquitin ligase of the SCF complex in atrophy-1 or *Musa1* [37] (Appendix A). Concomitantly, the expression of *apelin* was halved in C26- and LLC-bearing mice (Figure 2D,F) and cut to 30% in MCG101-carrying ones (Figure 2E). Instead, when injected in mice, tumors unable to cause BWL—such as the triple-negative breast cancer 4T1 [11]—neither *MuRF1* nor *apelin* expression in muscles appeared altered (Figure 2G,H) as well as the expression of *atrogin-1* and *Musa1* (Appendix A). As further control, we measured the protein content of MuRF1 in muscles of MCG101-carriers and 4T1-hosts and found increased MuRF1 in the former and unchanged in the latter, confirming Q-PCR data (Appendix A). Remarkably, in biopsies from rectus abdominis muscles of patients with diverse abdominal cancers [11], *apelin* expression was inversely correlated with *MuRF1* (Figure 2I).

Overall, apelin downregulation in muscles seems a hallmark of cancer cachexia regardless of the species (*Rattus Norvegicus*, *Mus musculus*, *Homo sapiens*) or the tumor type (hepatoma, sarcoma, lung or colon carcinomas).

### 3.2. Apelin Exerts Anti-Catabolic Action on Atrophying Myotubes

To better understand the effects of apelin on muscles, we moved from in vivo to in vitro myotubes. Muscles are composed of many different cells (satellite cells, fibers, endothelial cells, vessels, fibroadipogenic precursors, mesoangioblasts, etc.) [38], which may further complicate the picture.

We induced atrophy in four day-differentiated myotubes by exposing them for 24 h to 10 μM dexamethasone, a widely accepted atrophy inducer [39]. As expected, dexamethasone-treated myotubes became thinner (Figure 3A), and their fiber diameter was reduced by 30% (Appendix A). In line with this, the long-lived protein degradation rate of dexamethasone-treated myotubes increased by about 30% (Figure 3B), resulting in an overall reduction of about 25% of protein content (Figure 3C). In atrophying myotubes, when dexamethasone raised *MuRF1* expression by about 3-fold (Figure 3D) and *atrogin-1* by 3-fold (Appendix A), *apelin* expression was halved (Figure 3E), fully mimicking what occurred in vivo. Despite other cells present in muscles, muscle fibers were indeed less expressing *apelin* in vivo during cachexia.

Initially, we measured if myoblasts and myotubes from mice and humans expressed APJ by Q-PCR or WB and found that it was so, even if its expression during differentiation was slightly different in the two species (Appendix A). To address how myotubes react to apelin, we transfected myotubes/myoblasts with a plasmid expressing preproapelin (and verified by Q-PCR that apelin was expressed, Appendix A) or treated them with apelin peptides. Myotubes expressing preproapelin degrade long-lived proteins at rates similar to myotubes expressing an empty vector (pLKO) (Figure 3F). As expected, dexamethasone raised the rates of long-lived protein degradation, but this enhanced proteolysis was partially restrained by preproapelin overexpression (Figure 3F). When we transfected myoblasts with a reporter vector with the gene for *Firefly luciferase* (*FLuc*) under the control of *MuRF1* promoter, dexamethasone activated *MuRF1* promoter and *FLuc* expression increased almost 15-fold (Figure 3G). Preproapelin per se was unable to alter *MuRF1* expression, but attenuated the dexamethasone-induced *MuRF1* expression, exerting anti-catabolic action in atrophying cells (Figure 3G). Apelin 13 or 36 are the major peptide fragments of apelin binding APJ [16] and we found that apelin 13, unlike apelin 36, fully preserved myotube diameter from dexamethasone-induced atrophy (Figure 3H).

These data prompted us to test whether manipulating this pathway in muscle in vivo might also be beneficial against cancer cachexia.

### 3.3. Apelin 13 Carrying All the Amino Acids in D-Isomers Has Much Longer Half-Life In Vivo

Apelin 13 and its antagonist have already been used to treat various conditions in animals [21,40,41]. However, since peptide stability is a critical factor influencing their therapeutic effect, we measured the turnover of exogenous apelin in mouse plasma when given intraperitoneally. Mass spectrometry showed it to be around 3 h when all the peaks corresponding to apelin 13 disappeared (Figure 4A). We then replaced all the 13 amino acids of apelin with the corresponding amino acid in D-isomer, which is not degradable by mammalian systems [42], to extend its half-life. This apelin proved stable in mouse plasma for at least five days, when some peaks corresponding to degradation products started to appear and the major profile corresponding to apelin began to change, indicating post-translational modifications that may alter its bioavailability or APJ-binding ability (Figure 4B).

To test whether D-apelin still binds APJ, ultimately lowering cAMP levels, we transfected myoblasts with a reporter plasmid with the gene for *FLuc* under the control of cAMP-responsive elements (CRE). Then, we treated them with vehicle, L-apelin 13 or D-apelin 13, in combination with vehicle or forskolin, which induces intracellular cAMP [43]. As expected, forskolin induced the CRE-related bioluminescent signal more than 20-fold (Figure 4C). Since APJ was coupled to Gαi [44], both L- and D-apelin 13 reduced cAMP levels to similar extents, as indicated by the decrease in bioluminescent signal, showing that the L-to-D amino acid conversion does not affect apelin’s ability to alter the cAMP-signaling at all (Figure 4C).

### 3.4. Preproapelin or Apelin Given to Cancer-Bearing Mice Does Not Affect Muscle Size at All

To select the cachectic cell line more resistant to apelin for in vivo experiments, we treated the cancer cell lines able to cause cachexia (C26, LLC, MCG101) with these long-lived apelin peptides at increasing doses and times. Only supraphysiological doses of D-apelin 13 (250 μM at 24 h and 100 and 250 μM at 48 h) reduced the viability of most cancer cells (Appendix A).

We injected 3 × 10^10^ units of preproapelin-expressing AAV9 in the TA of mice previously injected with PBS or C26, and weighed muscles at sacrifice. There was no preservation of TA weight (Figure 5A); although apelin was properly overexpressed (Figure 5B), its inductions was able to restrain neither *MuRF1* expression (Figure 5C) nor that of *atrogin-**1* (Appendix A), as it did in vitro (Figure 3G). After having tested that D-apelin 13 has similar in vitro effects as L-apelin 13, we also injected D-apelin 13 in C26-bearing mice at doses and times that had previously assumed were not toxic in vivo. There was no body or MWL in tumor-free mice injected intraperitoneally (i.p.) with D-apelin 13 up to 1.5 μmol/Kg daily for 13 days (Figure 5D and data not shown). Again, apelin did not appear to have any effect on body or MWL (Figure 5E–H) or on tumor growth (Figure 5I), confirming that regardless of the route of administration, activation of this pathway does not preserve muscle size.

To ascertain whether D-apelin 13 injected intraperitoneally for 13 days daily to mice reached the muscles, we ran MALDI-TOF analyses on TA muscle. This confirmed that the peptides were present in the treated TA after the end of the treatment (Appendix A), definitely excluding any protective effect of apelin in cachectic muscles.

### 3.5. Muscles from Mice with Cancer Cachexia Exhibit Undesirable Apelin Resistance

In response to persistent stimulation, GPCRs become inactivated through a process named desensitization [45,46]. Since we could not see any effect on muscle size during cancer cachexia in mice treated with the agonist of APJ (Figure 5), we suspected that apelin could not exert any effects on muscles as in vitro, because APJ could be desensitized in cachectic muscles.

To address this issue, firstly we measured APJ protein content in muscles from healthy (i.e., PBS-injected) mice and cachectic ones bearing C26 or LLC tumors. Still, we found that the protein content of APJ was mostly stable in cachectic muscles (Appendix A). Then, we tested whether apelin activated AKT in muscles from PBS- and C26-injected mice (Figure 6). AKT tended to be more activated in TA from healthy mice previously injected with preproapelin-encoding AAV9, while AKT was not more phosphorylated (i.e., activated) in TA from apelin-treated C26 mice (Figure 6A,B and Appendix A). Similarly, 4EBP1 that acts downstream of AKT was found more phosphorylated (i.e., in a shape able to detach from eIF4E, which in turn promotes translation) in TA from healthy mice previously injected with preproapelin-encoding AAV9, but its phosphorylation in C26-mice was not altered by excess of apelin (Figure 6A,C and Appendix A). As expected, MuRF1 protein was upregulated in TA from C26-mice with respect to controls, but apelin failed to restrain this induction (Figure 6A–D and Appendix A). Similar results were found when muscles from mice treated with D-apelin 13 were analyzed (Appendix A). This indicated that APJ can be activated in healthy muscles and its unresponsiveness is a feature of only cachectic muscles, despite the ligand was long-lived (Figure 4B), detected in muscle (Appendix A), and the receptor still present on muscles in cachexia (Appendix A).

Finally, to understand whether APJ was rate-limiting during muscle wasting in cancer, we electroporated a plasmid encoding for APJ-GFP in TA muscle. The fluorescence tag enabled us to identify the fibers expressing the receptors and to compare their CSA with the adjacent negative ones. Consistently, APJ, but not apelin overexpression in muscles (Figure 5A,B), was sufficient to overcome the desensitization step by partially preserving the fibers from C26-induced atrophy (Figure 6E,F). Most importantly, the overexpression of APJ using plasmid electroporation resulted in mRNA levels of exogenous APJ that strongly anticorrelated with mRNA levels of *MuRF1* (Appendix A), oppositely to what occurs to levels of exogenous apelin, overexpressed by means of AAV that even displayed correlations with MuRF1 (Appendix A). This supported that the APJ desensitization can be circumvented by increasing levels of exogenous APJ, but can be worsened by apelin overexpression.

These data suggest that endogenous APJ seems somehow undesirably unresponsive in muscles during cachexia, nullifying any treatment involving agonists of this pathway.

### 3.6. Plasma Apelin Is Increased in Cachectic Cancer-Bearing Mice

The ability of apelin to protect myotubes from atrophy in vitro, but its failure to do so in vivo, prompted us to further explore the mechanism of apelin resistance during cancer cachexia in vivo.

Apelin was measured in plasma of cachectic mice bearing C26, MCG101, or LLC tumors. Surprisingly, apelin was increased more than three-fold in C26- and LLC-bearing male mice (Figure 7A,C) and by 25% in MCG101-carriers (Figure 7B), suggesting a persistent exposure of muscle APJ to high levels of apelin in cachectic mice. Hyperapelinemia was also seen in C26- and LLC-bearing female mice (Appendix A). Since the major apelin-producing organs, fat and muscles, are reduced during cancer cachexia, to understand whether the high levels of apelin in plasma could derive from tumors, we compared the protein content of apelin in tumors not causing cachexia, as 4T1, to tumors promoting cachexia, as C26 (Figure 7D,E and Appendix A). Notably, we found more apelin protein content in C26 than 4T1, supporting the increased plasma apelin as tumor-derived in cachectic conditions (Figure 7D,E and Appendix A).

Pancreatic cancer patients have a high incidence of cachexia (70–80%) [47]. To further confirm whether the high levels of apelin in plasma could come from tumors, we used the murine orthotopic pancreatic adenocarcinoma (PDAC) model FC1199 derived from tumors arisen in *LSL-Kras^G12D^*^/*+*^*;LSL-Trp53^R172H^*^/*+*^*;Pdx-1-Cre* mice described to induce cachexia in mice [48,49]. Since cachexia in this tumor model can be observed only in slow growing tumors, different numbers of FC1199 cells were transplanted orthotopically in the pancreas of C57BL/6 mice and *apelin* mRNA expression in tumors compared. Mice injected with 5 × 10^3^ tumor cells survive longer (31 days) and showed a significant loss of TA (cachectic mice) compared with mice injected with 1 × 10^4^ cells that survive only 26 days without signs of cachexia (Figure 7F). At sacrifice mRNA expression of apelin was significantly increased in tumors of cachectic mice presenting MWL compared with tumors of mice with no MWL, supporting apelin expression as a signature of cachexia-inducing tumors (Figure 7G).

These findings could suggest that the lower muscle levels of apelin might be a compensatory response to the increased circulating apelin secreted from tumors during cachexia and that apelin resistance may arise during cachexia.

## 4. Discussion

Cachexia is a an urgent medical need that worsens the prognosis of cancer patients. It has been estimated that worldwide around 9–14.5 million cancer patients suffer from cachexia every year [3]. Adipokines are fat-derived factors and include leptin, adiponectin, resistin, and visfatin for which a role in cancer cachexia has been suggested [50]. Apelin is also considered a member of this family and our data suggest overall that it would be potentially beneficial against cachexia.

Apelin has been shown to be useful against muscle wasting during aging, at least in animal models, facilitating muscle protein synthesis and mitochondriogenesis [21]. We believe we have unraveled another ability of apelin to exert anti-catabolic action on myotubes by restraining *MuRF1* expression and dexamethasone-induced proteolysis (Figure 3). This may have been the indirect result of AKT activation by apelin, but in apelin-treated myotubes we never found increased protein synthesis in radioactive tracer-based assays (data not shown) nor have we observed hypertrophied muscles in apelin-treated mice (Figure 5).

Mutants of constitutively active Gαi2 inhibit TNFα-induced muscle wasting, restraining MuRF1 inductions [51]. APJ, the receptor for apelin, is one of the GPCRs coupled to Gαi2 [44], which is involved in muscle hypertrophy [51], further linking apelin-APJ to counteraction of muscle atrophy. Reducing cAMP through apelin or other strategies would be useful in atrophied muscles because it has been shown that cAMP promotes the protein degradation by activating the proteasome through phosphorylation of proteasome subunits, such as Rpn6 by cAMP-dependent protein kinase [52]. So restraining muscle cAMP would translate into reduced proteasome activity and limited proteolysis. Since raising cAMP globally have undesirable side effects (i.e., nausea and emesis) that have prevented its use in patients, it would be better to pursue strategies to raise cAMP at muscle level.

### 4.1. Apelin Is Regulated in Opposite Ways in Muscle and Plasma during Cancer Cachexia

The downregulation of apelin expression in muscle and its increased circulation in blood may seem contradictory. Still, it can indeed be explained by an overproduction from other tissues as the tumoral one. Many tumors produce apelin that gives them growth advantages, so its inhibition might have antitumoral effects [53,54]. Our data support apelin as produced only by cachexia-promoting tumors (Figure 7), regardless of tumor types. In agreement with others who report increased plasma levels of apelin in human cancers, especially gastro-esophageal ones [55], we also describe the increased apelin in the blood of multiple cancer-bearing mouse models with cachexia (Figure 7).

The discrepancy between in vitro and in vivo effects of apelin may not reside on the different concentrations tested. In vitro we used 10 nM apelin as it has been used by others [16,24,25]. Indeed, we have calculated that 10 nM of preproapelin corresponds to about 62 ng/mL, since in plasma endogenous apelin concentration was 100–1000 pg/mL in mice (Figure 7A–C), if we assume that the same is in muscles, which is a highly vascularized tissue, an increase by 40 times that is the one we obtained by injection of preproapelin-encoding AAV9 (Figure 5B) would correspond to 4–40 ng/mL that has the same orders of magnitude than 62 ng/mL used in vitro. Although, this is based on the assumption that mRNA expression and protein secretion are highly correlated when, for most tissues, this does not always hold true [56,57,58,59].

Not all of the health-promoting effects of exercise may still act in cachexia. We believe that the beneficial effects of apelin during cachexia are hampered by APJ unresponsiveness in muscles. Desensitization of APJ has already been shown by Pope and coworkers, but in in vitro models [60]. Contrary to musclin, which may still be produced by cachectic muscle under exercise and exerts its anti-catabolic action [31], apelin that is equally increased during exercise [21,61,62] could not counteract cachexia because of receptor desensitization. Nonetheless, it remains to establish whether exercise counteracts APJ desensitization as it induces circulating apelin.

Since SDF1 and apelin are downregulated only in muscles from cachectic and tumor-bearing mice [11], it would be interesting to learn whether and how these two pathways crosstalk in muscles, like occurs for their receptors CXCR4 and APJ during blood vessel maturation [63]. Apelin downregulation and APJ desensitization in cachectic muscles may translate into reduced capillarization, possibly contributing to the oxidative-to-glycolytic metabolism switch seen in muscle with cachexia [64].

### 4.2. How Apelin and Glucocorticoids May Be Linked

To recapitulate in vitro cachexia, we treated myotubes with dexamethasone, a chemical compound resembling endogenous glucocorticoids. Glucocorticoids have been found elevated in the serum of C26-bearing mice [65], lung cancer-carrying mice [66], and tumor-bearing rats [67,68], as well as in the blood from cancer patients [69,70,71], even if their increase is not the only mechanism involved in cachexia. Many researchers use dexamethasone to induce atrophy in myotubes [31,39,72,73,74,75], because it can induce proteolysis, and anabolic factors, such as Insulin-like growth factor 1, oppose its action. We found that dexamethasone reduced the mRNA levels of apelin (Figure 3E) as opposed to its induction by insulin [76] in adipocytes. Since apelin was also reduced by dexamethasone in adipocytes [76], there seems to be some common regulation of apelin expression in fat and muscles. Shen and coworkers showed that in myotubes dexamethasone induced miRNA503* [77] that, through TargetScan7.1, we found able to restrain the expression of murine *apelin* (our unpublished observations). Further work is needed to see whether dexamethasone reduces apelin via miRNA and if the same occurs in adult muscles with cachexia or in dexamethasone-treated adipocytes.

### 4.3. Insulin Resistance and Apelin Resistance in Cancer Cachexia?

We believe that apelin resistance occurs during cachexia because (i) apelin is elevated in plasma of cachectic mice (Figure 7), (ii) while the apelin-mediated signaling can still be enhanced in healthy muscles, it seems refractory in cachectic muscles in vivo (Figure 6), and (iii) APJ replacement by muscle electroporation preserves fiber area from C26-related atrophy (Figure 6D,E). Apelin expression appears reduced in muscles of the elderly too [21] and in mice with insulin resistance [78], and insulin resistance arises in cancer patients [9] and in multiple cachectic models [8,9,79], including C26 [80]. At least in cells, the apelin expression seems under the control of insulin [76] and so, we may speculate that insulin insensitivity may account for apelin attenuation perhaps also in muscles. On the other hand, however, Yue et al. showed that apelin is required to maintain insulin sensitivity [15]. For these reasons, in our studies BALB/c mice have proven more suitable than C57BL6/J mice to study apelin resistance, since C57BL6/J mice are known to display impaired glucose intolerance that could have complicated the readouts [81].

## 5. Conclusions

Overall our data indicate that during C26 tumor induced-cachexia, apelin resistance may occur and worsen the progression of muscle wasting. Further studies are required to clarify whether during cachexia apelin resistance further worsens the insulin resistance status, feeding a deleterious vicious cycle that may further aggravate myopenia.

## Figures and Tables

**Figure 1 cancers-14-01814-f001:**
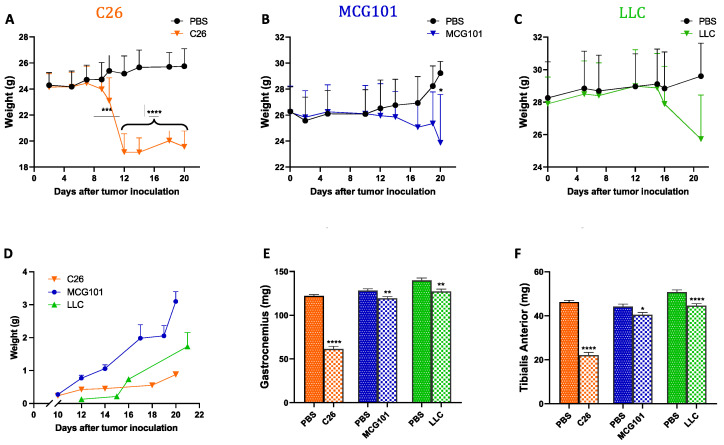
Different tumors induce body weight loss and muscle wasting in mice with different timing and tumor growth rate. In orange, 9–10 week-old male C26-bearing BALB/c mice start losing body weight already after 10 days from tumor injections and have lost 20% of body weight 12 days later, *n* = 9–10 (**A**). In blue, 9–10 week-old male MCG101-carrying C57BL6/J mice lose about 10% of their initial body weight after 20 days from tumor inoculation, *n* = 10–11 (**B**). In green, 9–10 week-old male LLC-bearing C57BL6/J mice present about 8% body wasting after 20 days, but this was not significant, *n* = 10 (**C**). Two-way ANOVA, * *p* ≤ 0.05, *** *p* ≤ 0.001, **** *p* ≤ 0.0001. In black, PBS-injected mice are indicated as controls (**A**–**C**). Tumor weights measured manually with a caliper over time are plotted (**D**). Ordinary one-way ANOVA, Tukey’s multiple comparisons test, not significant, *n* = 10–11. The weights of gastrocnemius (**E**) and tibialis anterior muscles (**F**) are reduced in all the tumor-bearing mice, but to different extents. In orange, C26 *n* = 18–20, in blue, MCG101 *n* = 20–22 and in green, LLC *n* = 20. Unpaired *t*-test, * *p* ≤ 0.05, ** *p* ≤ 0.01, **** *p* ≤ 0.0001.

**Figure 2 cancers-14-01814-f002:**
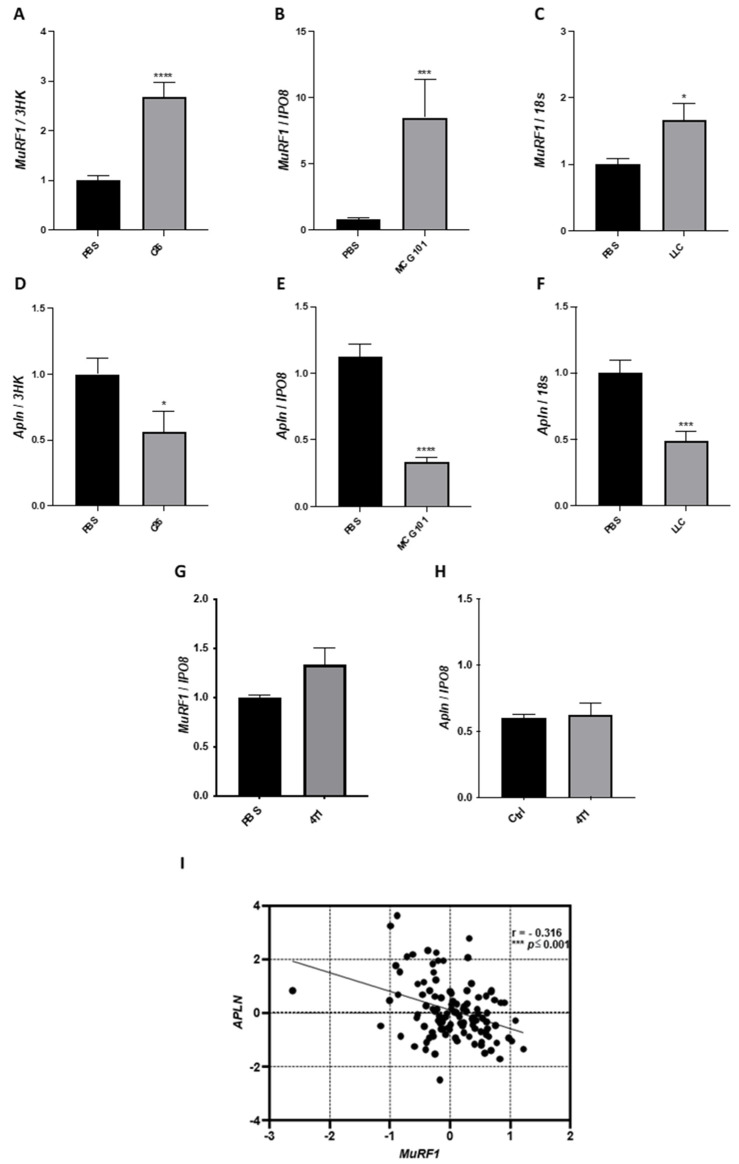
In the atrophied muscle of all the three animal models of cancer cachexia and cancer patients, *apelin* expression falls when *MuRF1* increases. Q-PCR shows that mRNA levels of the ubiquitin ligase *MuRF1* are highly induced in the TA of mice carrying the tumor C26 (**A**) or MCG101 (**B**) or LLC (**C**). Unpaired *t*-test, **** *p* ≤ 0.0001 (**A**); Mann–Whitney test, *** *p* ≤ 0.001 (**B**). Unpaired *t*-test, * *p* ≤ 0.05 (**C**), *n* = 7–10 (**A**–**C**). *Apelin* (*Apln*) expression is drastically decreased in TA from mice bearing C26 (**D**) or MCG101 (**E**) or LLC (**F**). Unpaired *t*-test, * *p* ≤ 0.05 (**D**), **** *p* ≤ 0.0001 (**E**), *** *p* ≤ 0.001 (**F**), *n* = 7–10. The mRNA levels of *MuRF1* are not induced in the TA of mice carrying the non-cachectic tumor 4T1 (**G**) and *apelin* expression is not altered in this tumor model (**H**). 3HK, three housekeeping genes (*TBP*, *IPO8,* and *Gusb*), *IPO8* or *Importin 8* and *18S* or *18S ribosomal RNA*, were used to normalize the data. Unpaired *t*-test, not significant, *n* = 5. In rectus abdominis muscles from cancer patients, as soon as *MuRF1* increases, *apelin* decreases (**I**). The scatter plot shows the inverse correlation between the expression of *apelin* and *MuRF1*. These expression data are log2-transformed. Spearman’s test, with correlation index (r) = −0.316, *** *p* ≤ 0.001, *n* = 115.

**Figure 3 cancers-14-01814-f003:**
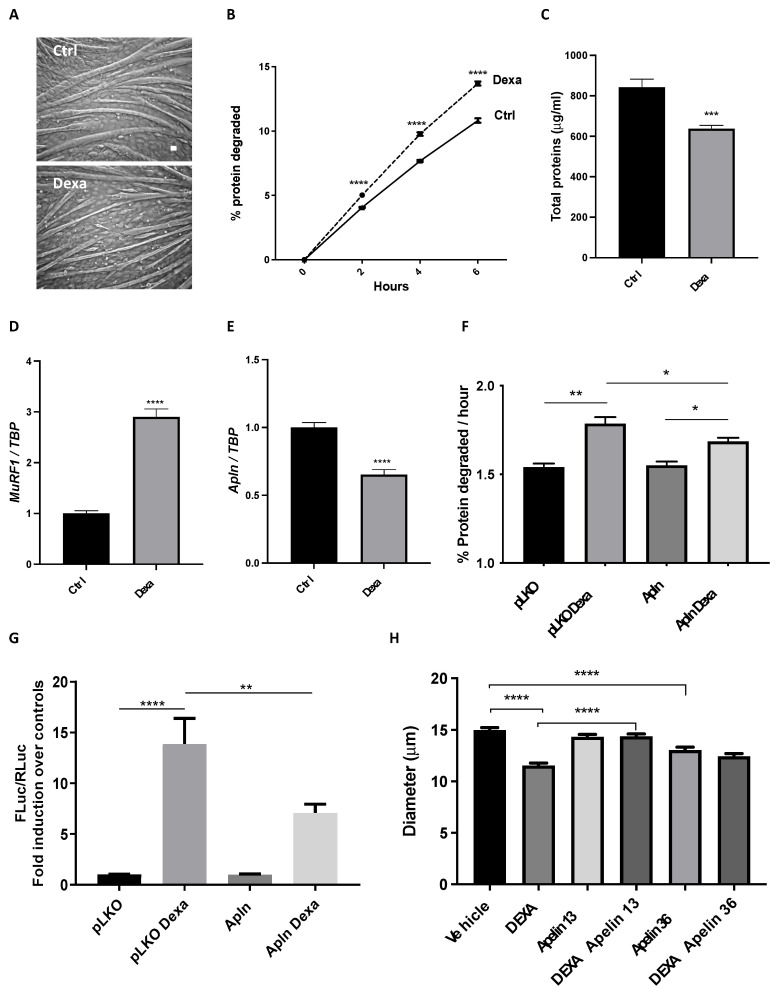
Apelin protects myotubes from dexamethasone-induced atrophy. Four day-differentiated myotubes were exposed for 24 h to vehicle (Ctrl) or 10 μM dexamethasone (Dexa) and representative brightfield images are shown. Scale bar, 15 μm (**A**). In radioactive-based assays, we measured long-lived protein degradation of myotubes treated with vehicle or 10 μM Dexa for 36 h; it was induced by 25% by Dexa (**B**). Multiple *t*-tests, **** *p* ≤ 0.0001, *n* = 6. Dexa-treated myotubes have lower total protein content than controls (Ctrl) (**C**). Unpaired *t*-test, *** *p* ≤ 0.001, *n* = 6. Like in vivo, Q-PCR shows that *MuRF1* expression is induced (**D**), while *apelin* expression is reduced (**E**) in myotubes treated for 24 h with 10 μM Dexa. *TBP* was used as housekeeping gene. Unpaired *t*-test, **** *p* ≤ 0.0001, *n* = 6. Four day-differentiated myotubes transfected with the empty vector pLKO, as control, or preproapelin-encoding plasmid for 24 h were treated with 10 μM Dexa for 24 h. Exogenous preproapelin partially restrained the Dexa-induced degradation of long-lived proteins (**F**). Ordinary one-way ANOVA, * *p* ≤ 0.05, *n* = 6, ** *p* ≤ 0.01. Myoblasts were transfected with a plasmid in which *Firefly luciferase* (FLuc) is under the control of a MuRF1 promoter, with another for *Renilla luciferase* (RLuc), under the control of thymidine kinase promoter to normalize the data (ratio 50:1) and with the empty vector pLKO, as control, or a preproapelin-expressing plasmid (Apln). After 24 h, myoblasts were treated with vehicle or with 10 μM Dexa for 24 h. Dexa-induced *MuRF1* expression was reduced in myoblasts that express exogenous preproapelin (**G**). Ordinary one-way ANOVA, ** *p* ≤ 0.01, **** *p* ≤ 0.0001, *n* = 12. Unlike apelin 36, apelin 13 fully protects myotubes from dexamethasone-induced atrophy. We measured in blind the diameter of four day-differentiated myotubes exposed for 24 h to vehicle or 10 μM Dexamethasone (DEXA) in combination with vehicle or 10 nM apelin peptides of different lengths. Dexamethasone reduced myotube diameter as expected by about 25%. Only the presence of apelin 13 fully protected them, while apelin 36 alone even reduced myotube diameters (**H**). Kruskal–Wallis test, **** *p* ≤ 0.0001, *n* = 261–297.

**Figure 4 cancers-14-01814-f004:**
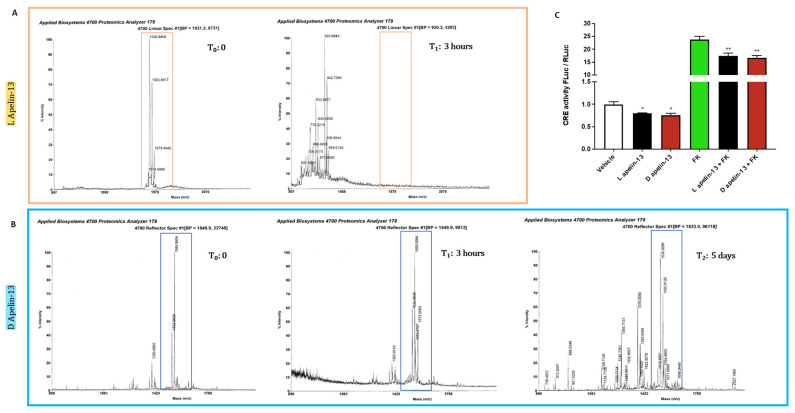
While L-apelin 13 is degraded in mouse serum in three hours, D-apelin 13 is stable for days. The peptide [Pyr1]-apelin 13 (MW 1533.81 g/mol) in L version (**A**) or D version of apelin 13 (**B**) were incubated at the same concentration in mouse serum at 37 °C. After different times from three hours to five days, a sample was analyzed using a MALDI-TOF mass spectrometer in linear mode. The molecular weights were compared with the original peptides at time 0. The highlighted peak that detects L-apelin 13 disappeared after 3 h (**A**), when additional peaks corresponding to lower molecular weights appear, indicating its fragmentation. Instead, D-apelin 13 was more stable in the mouse serum, because its peak (MW 1550 g/mol) persisted for five days (**B**). Myoblasts were transfected with a plasmid in which *Firefly luciferase* (FLuc) is under the control of cAMP responsive elements (CRE) and another for *Renilla luciferase* (RLuc) under the control of thymidine kinase promoter, to normalize the data. After 24 h, myoblasts were treated for 5 h with vehicle or 1 μM L-apelin 13 or D-apelin 13, in combination with vehicle or 50 μM forskolin and the ratio of FLuc/RLuc is plotted. L-apelin 13 and D-apelin 13 were equally able to reduce cAMP and forskolin-induced cAMP by about 25% in luciferase assays (**C**). Ordinary one-way ANOVA, * *p* ≤ 0.05, ** *p* ≤ 0.01, *n* = 6.

**Figure 5 cancers-14-01814-f005:**
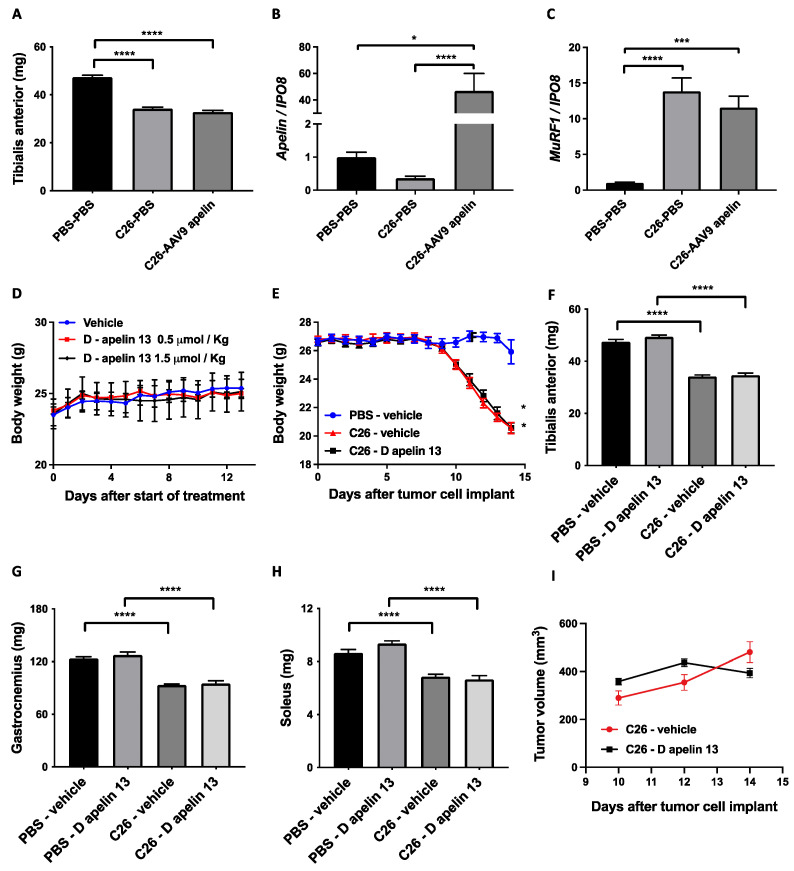
Intramuscular injection of preproapelin-expressing adenoassociated viruses or systemic treatment with D-apelin 13 do not protect muscles from cancer-induced wasting. Intramuscular injection of the preproapelin-expressing adenoassociated viruses type 9 (AAV9) does not preserve the TA from atrophy (**A**), even though the *apelin* gene is overexpressed in the muscle as shown by Q-PCR (**B**). C26-induced MuRF1 is not restrained by apelin overexpression (**C**). *IPO8* is used to normalize the data. Ordinary one-way ANOVA, Kruskal–Wallis test, * *p* ≤ 0.05, *** *p* = 0.0001, **** *p* ≤ 0.0001, *n* = 8–18. Vehicle (H_2_O) or 0.5 or 1.5 μmol/Kg D-apelin 13 were injected daily intraperitoneally in BALB/c mice and body weighs monitored over time. After 13 days, there were no differences in BWL among groups, excluding overt toxicity. Mouse body weights are plotted over time (**D**). One-way ANOVA for repeated measures, not significant, *n* = 3–4. C26 tumor cells subcutaneously injected in BALB/c mice cause BWL after 10 days from the injection, and D-apelin 13 does not preserve mouse body weight (**E**). One-way ANOVA for repeated measures, * *p* ≤ 0.05, *n* = 15. D-apelin 13 does not preserve muscles from atrophy induced by C26 cancer, as seen from TA (**F**), gastrocnemii (**G**) and solei (**H**) weighed at sacrifice, 14 days after tumor injection. Ordinary one-way ANOVA, **** *p* ≤ 0.0001, *n* = 15. Tumors were manually measured with a caliper. D-apelin 13 did not affect tumor growth rate (**I**). Multiple *t*-tests, not significant, *n* = 15.

**Figure 6 cancers-14-01814-f006:**
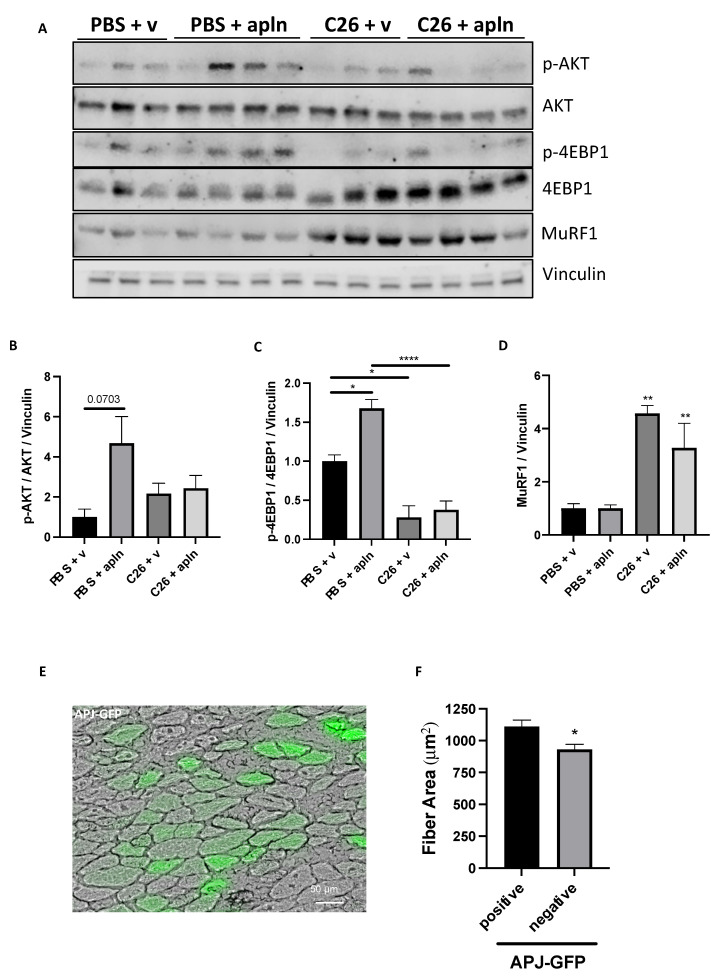
Muscles from mice with cancer cachexia display apelin resistance. Proteins from TA muscles of C26-carrying mice injected with preproapelin-expressing AAV9 were analyzed in Western blot for p-AKT, AKT, p-4EBP1, 4EBP1, MuRF1, and vinculin as loading control (**A**). Quantitations for p-AKT/AKT/vinculin (**B**), p-4EBP1/4EBP1/vinculin (**C**), and MuRF1/vinculin (**D**) are shown. Ordinary one-way ANOVA, Tukey’s multiple comparison test, * *p* ≤ 0.05, ** *p* ≤ 0.01, **** *p* ≤ 0.0001, *n* = 3–4. The APJ-GFP-encoding plasmid was electroporated for 14 days in mice that were injected with C26 tumor cells the day after. A representative image is shown (**E**). Scale bar, 50 μm. At sacrifice, TA were dissected, frozen, and cut and the mean fiber CSA of APJ-GFP-expressing fibers was found bigger than that of adjacent non-expressing ones (**F**). Unpaired *t*-test, Mann–Whitney test, * *p* ≤ 0.05, seven mice and 89 fibers.

**Figure 7 cancers-14-01814-f007:**
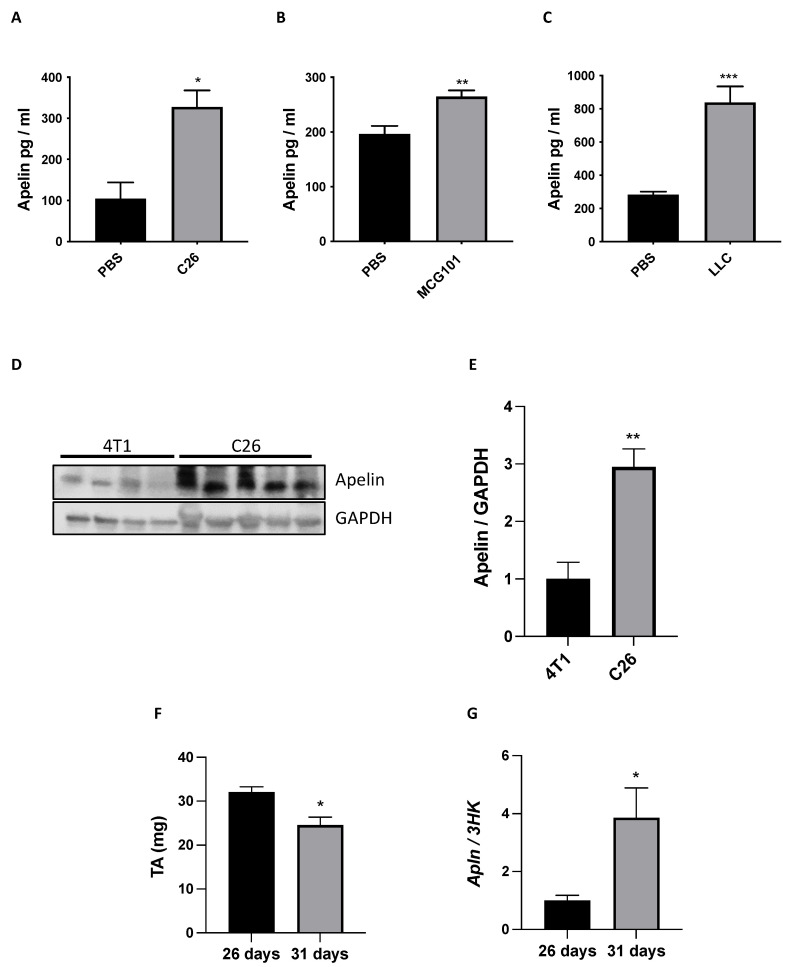
Plasma apelin is increased in cachectic cancer-bearing mice and it seems tumor-derived. ELISA was done on the plasma of mice with the cachectic tumors C26 (**A**), MCG101 (**B**) or LLC (**C**). Apelin was high in plasma of all three cancer types-bearing animals. Mann–Whitney test, * *p* ≤ 0.05, ** *p* ≤ 0.01, *** *p* ≤ 0.001. PBS, C26, *n* = 4–5; PBS, MCG101, *n* = 10–14; PBS, LLC, *n* = 6–10. Proteins from 4T1 or C26 tumors excised from mice were analyzed in Western blot for apelin and GAPDH as loading control (**D**). Quantitation for apelin/GAPDH is shown (**E**). Unpaired *t*-test, ** *p* ≤ 0.01, *n* = 4–5. TA weight of non-cachectic and cachectic pancreatic cancer bearing-mice is shown (**F**), Unpaired *t*-Test, * *p* ≤ 0.05, *n* = 5–6. Q-PCR showed that the tumoral mRNA levels of apelin were higher in cachectic than in non-cachectic mice bearing pancreatic tumors and sacrificed, respectively, 31 vs. 26 days after tumor transplantation (**G**). Unpaired *t*-test, * *p* ≤ 0.05, *n* = 5–6.

## Data Availability

Array Express repository (ID E-MTAB-8457) includes data on apelin expression shown in the text of Results section, Section 3.1. The microarray data sets (see Figure 2I) are deposited in the US National Center for Biotechnology Information Gene Expression Omnibus (GEO) and are accessible through GEO series accession number GSE41726.

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
