# Peer review of "Apelin Resistance Contributes to Muscle Loss during Cancer Cachexia in Mice"

_cancers, 2022, doi:10.3390/cancers14071814_

Round 1
Reviewer 1 Report
The authors improved the manuscript.
However the authors did not provide convincing answers to all the comments. The answers even raise important questions about this study.
1) Figure 2 I, about correlation of apelin/murf1 mRNA: it is still unclear how the graphic is working. Each mRNA expression is expressed to a scale with positive and negative values. Is it the fold change expression between cachectic and non cachectic patients?
More importantly, the authors report 137 patients with characteristics described in Bédard et al. First, the initial study concerned only 115 patients. Where did the added 22 patients come from? Secondly, in the patient characteristics it is noted that 34 of them have benign neoplasm. Are these 34 patients included in the correlation?
2) On the preclinical model, the author provides new data with a PDAC model. The reference cited to justify the model is useless to determine if this model is developing cancer cachexia: muscle loss? reduction in food intake? increase in inflammation....I assumed that this model can be a model of PDAC induced cachexia, but the authors have to prove it.
3) About PDAC patients, I'm still considering that comparing cancer patients (with high occurrence of cachexia) to healthy subjects is absolutely scientifically incorrect. These data are biased, since the authors compared cancer + cachexia vs nothing... These data must be removed or the authors must have PDAC cancer patients without cachexia as control.
4) How the authors explained that MuRF1 and apelin are correlated in cancer patients but not in preclinical model?
Author Response
The authors improved the manuscript.
However the authors did not provide convincing answers to all the comments. The answers even raise important questions about this study.
1) Figure 2 I, about correlation of apelin/murf1 mRNA: it is still unclear how the graphic is working. Each mRNA expression is expressed to a scale with positive and negative values. Is it the fold change expression between cachectic and non-cachectic patients?
Each dot represents an individual with cancer whose biopsy was analyzed for muscle expression of various genes. The data that do not represent the fold change between cachectic and non-cachectic patients are log transformed that is why some values are negative. We have added a sentence in the legend to make it clearer.
More importantly, the authors report 137 patients with characteristics described in Bédard et al. First, the initial study concerned only 115 patients. Where did the added 22 patients come from? Secondly, in the patient characteristics it is noted that 34 of them have benign neoplasm. Are these 34 patients included in the correlation?
The reviewer is absolutely right: we replaced the figure with the right one showing 115 patients whose characteristics are present in the paper (Bedard et al., 2015), including 34 patients with benign tumors. We have repeated the analyses deleting such 34 patients and still found statistical significant anticorrelation between apelin and MuRF1 (r = 0,261, p value = 0,0112, Spearman’s Test). Since we and others have already published gene expression data using all the 115 patients (Bedard et al., 2015 and Martinelli et al., 2016), we consider worth including all of them in the new Figure 2I.
2) On the preclinical model, the author provides new data with a PDAC model. The reference cited to justify the model is useless to determine if this model is developing cancer cachexia: muscle loss? reduction in food intake? increase in inflammation....I assumed that this model can be a model of PDAC induced cachexia, but the authors have to prove it.
We are grateful to the reviewer to have raised this issue. In the revised version, we provide two new references for this model PMID 33851955 and 28730707. This model causes muscle weight loss as already indicated in Figure 7D. They do not suffer from anorexia (personal communication).
3) About PDAC patients, I'm still considering that comparing cancer patients (with high occurrence of cachexia) to healthy subjects is absolutely scientifically incorrect. These data are biased, since the authors compared cancer + cachexia vs nothing... These data must be removed or the authors must have PDAC cancer patients without cachexia as control.
Because of the reviewer’s criticism on this note, we deleted this figure from the new submitted version, please see novel Figure 7.
4) How the authors explained that MuRF1 and apelin are correlated in cancer patients but not in preclinical model?
We believe that the such anti-correlation is poor and arises from the numerous n of patients (n = 134 now 115) that were much more than those samples analyzed in mice (n = 30, where MuRF1 and apelin do not correlate to each other (r = 0,2489, P = 0,1846, Spearman’s Test).
Reviewer 2 Report
The revised manuscript has addressed the issues that the paper had in its first form.
Author Response
The revised manuscript has addressed the issues that the paper had in its first form.
We are glad the reviewer appreciated all the effort we put to improve our work.
Reviewer 3 Report
Cecconi et al. have thoroughly revised the manuscript as well as provided a detailed response to my comments and suggestions. While I appreciate the modifications of the manuscript done in response to my suggestions, I also accept most explanations and arguments, which were provided to explain why my suggestions were not followed.
Comments:
- The approach to diagnosing cachexia has been clarified (TA weight loss); however, what was the exact criterion for cachexia (i.e. what was the threshold for the extent of TA weight loss to diagnose cachexia – 5%, 10%, 15%...?).
- To what extent did physical inactivity in tumour-bearing mice contribute to the loss of skeletal muscle mass?
- Figure 2: My comment regarding the endogenous controls was meant to highlight that while gene expression in cachectic models were evaluated with several endogenous controls (the C26 model even with three - 3HK), it was evaluated only with one endogenous control (IPO8) in the non-cachectic model 4T1. I think the data would be strengthened if additional controls were used to evaluate MuRF1 an apelin expression in this model.
- The discrepancy between in vitro and in vivo that I was referring to is as follows: experiments in C2C12 myotubes show that apelin opposes dexamethasone-induced proteolysis and degradation (Figure 3). However, in vivo there was no such protection (Figure 5). While glucocorticoids may contribute to cachexia, they are certainly not the only underlying mechanism. Therefore, while apelin treatment may protect against glucocorticoid-induced atrophy (in vitro or possibly in vivo), it does not seem to be effective against atrophy or cachexia produced by cancer-related factors in vivo.
- I still think that treatment of C2C12 cells with D-apelin in the presence or absence of dexamethasone to assess protein degradation or myotube diameter would be very valuable, but I accept that cAMP data indirectly suggest that L- and D-apelin possibly have the same effect on dexamethasone-induced atrophy.
- To what extent does skeletal muscle contribute to the plasma apelin concentrations? Plasma apelin levels are increased in cancer-bearing mice, while its expression in skeletal muscle is reduced. This would seem to suggest that skeletal muscle is not contributing the majority of circulating apelin. Also, if muscle and blood concentration of apelin were similar one would expect a marked increase in circulating apelin concentrations in AAV9 mice. So, is it known what was the apelin concentration in mice treated with D-apelin 13 and AAV9 mice (Figure 5)? This would be an important piece of information.
- I accept that local concentrations of apelin may have been high in skeletal muscle; however, as regards quantification, I think it is difficult to judge about that based just on the plasma concentration of apelin because its expression in skeletal muscle and its plasma concentration do not seem to correlate well. So, is it really possible to say that a 40-fold increase in mRNA expression leads to a 40-fold increase in the protein concentration? Was plasma concentration for apelin in AAV mice markedly increased (please see the above question #6)?
- If the correlation index stands for the correlation coefficient I think it would be better to use the standard terminology/nomenclature (e.g. correlation coefficient, r instead of R).
- “Similarly, 4EBP1 that acts downstream of AKT was found more phosphorylated (i.e., activated)…” Phosphorylation of 4EBP1 disrupts its interaction with eIF4E (rather than activating 4EBP1), which in turn promotes translation. The statement should be corrected.
- A few minor corrections need to be made, such as “please add REF PMID…”, “(PMID: 25071331)”,…
Author Response
Cecconi et al. have thoroughly revised the manuscript as well as provided a detailed response to my comments and suggestions. While I appreciate the modifications of the manuscript done in response to my suggestions, I also accept most explanations and arguments, which were provided to explain why my suggestions were not followed.
Comments:
- The approach to diagnosing cachexia has been clarified (TA weight loss); however, what was the exact criterion for cachexia (i.e. what was the threshold for the extent of TA weight loss to diagnose cachexia – 5%, 10%, 15%...?).
We did not set any arbitrary threshold in advance to define cachexia in terms of loss of TA weight, we only compared cancer-bearing mice with to cancer-free ones and apply statistical tests to unravel meaningful differences.
- To what extent did physical inactivity in tumour-bearing mice contribute to the loss of skeletal muscle mass?
The reviewer raises an interesting issue. Unfortunately, we did not measure physical inactivity of mice in our experiments. We will take into consideration this point in future experiments. However, the in vivo tumors were limited in size (especially for C26 mice where tumors reached a maximum size of less than 1 gr, but less true for MCG101- or LLC- carriers with tumors of about 3 or 2 gr at sacrifice, respectively, please see Figure 1D), so that we may assume these tumors may not impact on the movement of mice, but we cannot exclude that cage inactivity may occur due to advanced cachexia. At sight, they move less close to the day of the sacrifice. Inactivity although was one of the criteria of sacrifice, we added a sentence in this regard (see page 5 of the new version).
- Figure 2: My comment regarding the endogenous controls was meant to highlight that while gene expression in cachectic models were evaluated with several endogenous controls (the C26 model even with three - 3HK), it was evaluated only with one endogenous control (IPO8) in the non-cachectic model 4T1. I think the data would be strengthened if additional controls were used to evaluate MuRF1 an apelin expression in this model.
We thank the reviewer to have clarified this point, we checked our non -cachectic 4T1 model with other housekeeping genes, without finding any increase of the MuRF1 and atrogin1 or decrease or apelin. Please see attached figure that we share with this reviewer only. We do not consider appropriate to add this piece of information in the supplementary section, that already contains 14 figures.
- The discrepancy between in vitro and in vivo that I was referring to is as follows: experiments in C2C12 myotubes show that apelin opposes dexamethasone-induced proteolysis and degradation (Figure 3). However, in vivo there was no such protection (Figure 5). While glucocorticoids may contribute to cachexia, they are certainly not the only underlying mechanism. Therefore, while apelin treatment may protect against glucocorticoid-induced atrophy (in vitro or possibly in vivo), it does not seem to be effective against atrophy or cachexia produced by cancer-related factors in vivo.
We understand the point of the reviewer and have added a sentence in the Discussion to reduce the tone of our findings, see page 18 of the new version.
- I still think that treatment of C2C12 cells with D-apelin in the presence or absence of dexamethasone to assess protein degradation or myotube diameter would be very valuable, but I accept that cAMP data indirectly suggest that L- and D-apelin possibly have the same effect on dexamethasone-induced atrophy.
We are glad that the reviewer accepted our argument to support our conclusions.
- To what extent does skeletal muscle contribute to the plasma apelin concentrations? Plasma apelin levels are increased in cancer-bearing mice, while its expression in skeletal muscle is reduced. This would seem to suggest that skeletal muscle is not contributing the majority of circulating apelin. Also, if muscle and blood concentration of apelin were similar one would expect a marked increase in circulating apelin concentrations in AAV9 mice. So, is it known what was the apelin concentration in mice treated with D-apelin 13 and AAV9 mice (Figure 5)?This would be an important piece of information.
As we showed in the Figure 7 of the re-submitted version, we believe the apelin increases may be tumor-derived. D-Apelin 13 cannot be measured through ELISA because it is not recognized by commercial antibodies able only to recognize the L forms of peptides. With regard to plasma of mice injected in TA with preproapelin-encoding AAV9, we measured it through MALDI-TOF, but failed to see a systemic increase of apelin with respect to proper controls. Nonetheless, we were not surprised by this because the overexpression of apelin was local, in other words only in one muscle of the leg, it is unlikely that this may result in detectable increase of apelin systemically in plasma.
- I accept that local concentrations of apelin may have been high in skeletal muscle; however, as regards quantification, I think it is difficult to judge about that based just on the plasma concentration of apelin because its expression in skeletal muscle and its plasma concentration do not seem to correlate well. So, is it really possible to say that a 40-fold increase in mRNA expression leads to a 40-fold increase in the protein concentration? Was plasma concentration for apelin in AAV mice markedly increased (please see the above question #6)?
The reviewer is right the correlation between tissue mRNA expression and tissue protein is poor. It has been estimated in general with a “r” around 0,2-0.6 (PMID 23436904, 15075390, 27284200, 8863742) as well as resident plasma protein concentration poorly correlates with mRNA expression, for example in liver. We have added sentences on this topic in the Discussion, please see page 17 of the new version. We have already measured apelin in the blood of AAV9-injected mice, but we failed to see any increase, see reply to the point above.
- If the correlation index stands for the correlation coefficient I think it would be better to use the standard terminology/nomenclature (e.g. correlation coefficient, r instead of R).
We now understand the point of the reviewer and amended the text and figure accordingly.
- “Similarly, 4EBP1 that acts downstream of AKT was found more phosphorylated (i.e., activated)…” Phosphorylation of 4EBP1 disrupts its interaction with eIF4E (rather than activating 4EBP1), which in turn promotes translation. The statement should be corrected.
We are grateful to the reviewer to have found this mistake in our paper and corrected the text accordingly, see page 14.
- A few minor corrections need to be made, such as “please add REF PMID…”, “(PMID: 25071331)”,…
Again, we thank the reviewer, but since the bibliography was not formattable from the version we downloaded from the MDPI website, we kindly asked assistance to the MDPI staff to insert/format the references. That is why also in this new version, s/he will see these sentences.

Round 2
Reviewer 1 Report
As authors don't know the nutritional status of PDAC patients, there is absolutely no point to have these data in an manuscript investigating cancer cachexia. Comparing unknown nutritional status PDAC patients to healthy subjects indicates only cancer effect. PDAC is associated with an increase in plasma apelin concentration. Authors can't write " The roleTo verify its potential clinical value as a biomarker of plasma cachexia, apelin in APJ desensitization was further supported by data obtained from PDAC patients." No, the data absolutely do not support that since i) there isn't any information on nutritional status of PDAC patients and ii) PDAC were compared to healthy subjects. Authors must remove figure 7F and G from the manuscript.Figure 7 legend is a real mess.
Author Response
We agree with the reviewer and have already deleted that two Figures 7F and 7G from our manuscript. Probably, the track and changes modes did not allow the reviewer to see this. We hope in this shape our paper is suitable for publication in Cancers.
Reviewer 3 Report
Cecconi et al. have further improved the manuscript and I appreciate the PCR results that were provided for me. I congratulate the authors for the extent of work performed for this interesting study. I have just a two remaining comments:
- Cachexia in mice was not diagnosed by the extent of skeletal muscle of body weight loss, but by the presence of the cachectic tumour as opposed to non-cachectic tumour. For the sake of clarity for the readers I suggest that the authors indicate this fact in the methods. The current statement in the methods (“Mice were considered cachectic when exhibiting significant muscle loss with respect to cancer-free mice…”) implies that there may have been a predefined level of muscle loss to establish cachexia.
- The authors should carefully check the text. There seem to be typing errors, which were introduced during revision, e.g.: “The roleTo verify its potential clinical value as a biomarker of plasmacachexia, apein APJ desensitization was further supported by data obtained from PDAC patients.”.
Author Response
We have amended all the corrections proposed by the reviewer. Briefly, on page 5, highlighted in yellow, we have added a sentence to better clarify how cachectic mice were defined. We hope in this shape our paper is suitable for publication in Cancers.
Round 3
Reviewer 1 Report
The reviewer is really sorry, but as indidated before the preclinical model of PDAC induced by FC1199 cell injection isn't described as a model of cachexia. In the 3 articles cited by the authors, one is about FC1199 model, w/o any characterization of cachexia, and the 2 other articles are about PKC model, which is whell described as model of cachexia. Reduction in TA mass is not enough to characterize cancer cachexia. Are mice having reduction in food intake? body weight? adipose tissue loss? How the authors determined 26 and 31 days for sacrifice? What is the non cachectic group: mice with FC1199 that were not cachectics, or healthy mice? At these days after induction how was the nutritional status of mice? Because if you compare dying mice (extreme cachexia) to healthy mice, 100% chance that you get a large modification of any factor that is measured. Without details for characterization of FC1199 model as model of PDAC induced cachexia, authors must remove these data from the manuscript. It should not be difficult to remove this data, as it does not add anything to the article.
Title of figure 7 need to be updated
Author Response
We believe that data on apelin mRNA expression in FC1199 tumors are fundamental to prove that high levels of apelin in plasma could derive from tumors. As now stated in the text, FC1199 cell line derives from tumors arisen in LSL-KrasG12D/+;LSL-Trp53R172H/+;Pdx-1-Cremice a GEM model of PDAC already described to induce cachexia in mice. In addition to the references previously mentioned a recent paper describing the cachectic effect caused by FC1199 has been cited. This reference has been added at page 16 (PMID: 33824339.
Text has now been modified (page 16) to clarify the points arisen by the reviewer:
to understand whether the high levels of apelin in plasma could derive from tumors, we used the murine orthotopic pancreatic adenocarcinoma (PDAC) model FC1199 derived from tumors arisen in LSL-KrasG12D/+;LSL-Trp53R172H/+;Pdx-1-Cremice described to induce cachexia in mice (PMID: 33824339; 33851955). FC1199 cells (1x104and 5x103, respectively) were transplanted orthotopically in the pancreas of C57BL/6 mice and apelin mRNA expression was measured in FC1199 tumors. Mice injected with 5x103 tumor cells survive longer (31 days) and showed a significant loss of TA (cachectic mice) compared with mice injected with 1x104tumor cells that survive only 26 days (non-cachectic mice) (Figure 7D). At sacrifice mRNA expression of apelin was significantly increased in tumors of cachectic mice presenting MWL compared with tumors of mice with no MWL, supporting apelin expression as a signature of cachexia-inducing tumors (Figure 7E).
Notably, these cachectic mice showed an increasing even non-significant trend of expression of Murf1 and Atrogin1 in their muscles (please see attached file).

Round 4
Reviewer 1 Report
Thanks to the authors for clarifying some issues about the PDAC model by providing important details.
However, it is absolutely not clear why the authors used the PDAC model to "understand whether the high levels of apelin in plasma could derive from tumors". C26 and LLC models also have a significant reduction in TA weight. The authors must clarify their reasonning for the readers and justify why they didn't measure tumor mRNA expression of apelin in these models.
figure 7E: "Q-PCR showed that the tumoral mRNA levels of apelin were higher in cachectic than in non-cachectic mice bearing pancreatic tumors and sacrificed 31 vs 26 days after tumor transplantation".
First, the authors must explain why they used different times for sacrifice? and secondly the figure 7E shows only mRNA level between cachectic and non cachectic mice, "31 vs 26 days after transplantation" is not reflected on the graph.
Title of figure 7 still needs to be corrected: Figure 7. "Plasma apelin is increased in cachectic cancer-bearing mice and patients".
Author Response
Thanks to the authors for clarifying some issues about the PDAC model by providing important details.
However, it is absolutely not clear why the authors used the PDAC model to "understand whether the high levels of apelin in plasma could derive from tumors". C26 and LLC models also have a significant reduction in TA weight. The authors must clarify their reasoning for the readers and justify why they didn't measure tumor mRNA expression of apelin in these models.
The reviewer is right on this point, so we added new datasets showing protein levels of apelin that seemed increased in tumors causing cachexia as C26, with respect to those non-causing cachexia as 4T1, please see novel Figure 7, panel D and E of the novel version.
figure 7E: "Q-PCR showed that the tumoral mRNA levels of apelin were higher in cachectic than in non-cachectic mice bearing pancreatic tumors and sacrificed 31 vs 26 days after tumor transplantation".
First, the authors must explain why they used different times for sacrifice? and secondly the figure 7E shows only mRNA level between cachectic and non-cachectic mice, "31 vs 26 days after transplantation" is not reflected on the graph.
The reason of injecting different number of FC1199 cells is now explained in the text on page 16 of the newer version:
Since cachexia in this tumor model can be observed only in slow growing tumors, different number of FC1199 cells were transplanted orthotopically in the pancreas of C57BL/6 mice and apelin mRNA expression in tumors compared. Mice injected with 5x103tumor cells survive longer (31 days) and showed a significant loss of TA (cachectic mice) compared with mice injected with 1x104cells that survive only 26 days without sign of cachexia (Figure 7F). At sacrifice mRNA expression of apelin was significantly increased in tumors of cachectic mice presenting MWL compared with tumors of mice with no MWL, supporting apelin expression as a signature of cachexia-inducing tumors (Figure 7G).
We agree with the reviewer that 26 and 31 days shall replace the non-cachectic and cachectic legends of these figures, respectively, so we did this accordingly.
Title of figure 7 still needs to be corrected: Figure 7. "Plasma apelin is increased in cachectic cancer-bearing mice and patients".
We are grateful to the reviewer and have amended this, accordingly.
